# SARS-CoV-2 Infection during Pregnancy Followed by Thalamic Neonatal Stroke—Case Report

**DOI:** 10.3390/children10060958

**Published:** 2023-05-27

**Authors:** Diana Iulia Vasilescu, Ana Maria Rosoga, Sorin Vasilescu, Ion Dragomir, Vlad Dima, Adriana Mihaela Dan, Monica Mihaela Cirstoiu

**Affiliations:** 1Doctoral School, Carol Davila University of Medicine and Pharmacy, 050474 Bucharest, Romania; 2Faculty of Medicine, Carol Davila University of Medicineand Pharmacy, 020956 Bucharest, Romania; 3Department of Neonatology, Emergency University Hospital, 050098 Bucharest, Romania; 4Department of Obstetrics and Gynecology, Emergency University Hospital, 050098 Bucharest, Romania; 5Department of Neonatology, Marie Curie Children Hospital, 041451 Bucharest, Romania; 6Department of Neonatology, Filantropia Clinical Hospital, 011171 Bucharest, Romania

**Keywords:** SARS-CoV-2, stroke, apnea, preterm, thalamic ischemic process, preeclampsia

## Abstract

A neonatal stroke is a cerebrovascular process caused by interruption of cerebral blood flow that occurs with an incidence between 1 per 1600 and 1 per 2660 live births. Relative higher incidence in the neonatal period compared to later childhood is favored by the hypercoagulability state of the mother, mechanical stress during delivery, transient right to left intracardiac shunt, high hematocrit, blood viscosity, and risk of dehydration during the first few days of life. The exact cause of a neonatal stroke remains unclear in many cases. About 80% of neonatal strokes are due to arterial ischemic events involving the middle cerebral artery. Typical clinical manifestations in a neonatal stroke are usually seizures that appear immediately after birth or after several days of life, but many of the cases may remain asymptomatic. We present the case of a late preterm infant diagnosed with a thalamic stroke on the fifth day of life with no clinical signs except for repeated episodes of apnea. The anamnesis and clinical context, in this case, revealed a SARS-CoV-2 infection in late pregnancy and early bacterial neonatal sepsis. Early identification of a perinatal stroke and increasing awareness of physicians about this condition in the neonatal period have paramount importance to reduce developmental postischemic damage.

## 1. Introduction

A neonatal stroke is defined as an acute cerebrovascular process caused by interruption of venous or arterial cerebral blood flow that leads to severe modifications in white or gray matter architecture [1]. It can be divided into sinovenous thrombosis and an arterial ischemic stroke [2], can be a localized or multifocal process that usually occurs in term or late preterm infants, and is diagnosed using neuroimaging techniques during the first 28 days of life [3,4]. Existing evidence has demonstrated that the incidence of a neonatal stroke is estimated at 1 per 1600 to 1 per 2660 live births [5] and is 17 times more common in neonates than in children and adolescents [6]. A proportion of 80% of neonatal strokes is due to arterial ischemic events, usually involving the middle cerebral artery and 20% are demonstrated to be caused by cerebral venous sinus thrombosis or a brain hemorrhage [7]. The superior sagittal sinus and the lateral sinuses are more commonly involved. In contrast, a study conducted by Berfelo et al. revealed that the straight sinus was most affected and an associated thalamic hemorrhage was common [8].

With multifactorial, complex etiology, the cause of a neonatal stroke is often difficult to recognize and may remain unclear in many cases. The most significant maternal risk factors are chorioamnionitis, thrombophilia, coagulopathy, preeclampsia, hypoglycemia, and drug exposure (cocaine), combined with perinatal inflammation, intrauterine growth restriction, perinatal asphyxia, and congenital cardiac diseases [2,9].

Other causes of a perinatal arterial ischemic stroke may be a cerebral embolism explained by two pathogenetic theories: a placenta–embolic hypothesis and a traumatic one [10,11,12]. The first theory asserts that an embolus from the placenta passes through the foramen ovale and reaches the brain, where the injury occurs. The traumatic theory considers that during delivery there might be an injury that involves the cervical–cerebral arteries, which leads to the formation of an embolus.

There are also genetic conditions that increase the risk of a perinatal stroke. The recent development of molecular genetic methods has made possible the identification of genetic disorders, both single and multiple gene mutations that are associated with increased risk of stroke in the pediatric age; most studied are several common polymorphisms in genes associated with thrombophilia [13]. In a population-based case control study undertaken between 1997 and 2002 at Kaiser Permanente Northern California, 13 infants with a perinatal arterial ischemic stroke were identified [14]. These patients were genotyped for a polymorphism in nine genes involved in inflammation, thrombosis, or lipid metabolism. These genes were TNF alpha-308, IL6, lymphotoxin A, factor V Leiden, MTHFR 1298 and 667, prothrombin 20210, and apolipoprotein E Ɛ2 and Ɛ4 alleles, which had been previously linked with strokes. Comparing genotype frequencies in the case and control individuals, it has been suggested that the apolipoprotein E polymorphism may confer genetic susceptibility for an arterial ischemic stroke. Proinflammatory and prothrombotic polymorphisms were not associated with increased risk in this study [14].

Central nervous system infections may also increase the risk of a perinatal stroke. Meningeal inflammation can promote a prothrombotic process in the cerebral vasculature, resulting in a venous or arterial infarction [15]. In a study on 166 children with meningitis, ischemic injury was identified in 10% of the patients [16]. Patterns of ischemic injury detected through MRI in these patients were subdivided into three categories: multiple punctate lesions in the basal ganglia, focal or diffuse cerebral infarcts, and focal subcortical or periventricular white matter lesions [14,17].

Among other maternal conditions that can increase the risk of a perinatal stroke, SARS-CoV-2 infection in pregnancy may be taken into consideration. Specific findings regarding a cerebral stroke in neonates born to mothers with SARS-CoV-2 infection during pregnancy are limited, although studies have shown a possible link between the viral infection and stroke in older children [18,19,20]. It is demonstrated that SARS-CoV-2 is associated with neurological manifestations in adults, raising the hypothesis of a direct neuropathogenicity of the virus, besides the pro-coagulant status that it may induce. In a meta-analysis published by Maury et al. in 2021, 1.3–4.7% of patients with SARS-CoV-2 infection were diagnosed also with acute cerebrovascular diseases, including ischemic strokes [21]. Responsible mechanisms for these multiple types of CNS injuries in these patients were considered to be a post-infectious response, septic-associated encephalopathies, coagulopathy, or endotheliitis [21].

Typical clinical manifestations in a neonatal stroke are usually seizures that can appear immediately after birth or later, after several days of life. Apnea, hemiparesis, lethargy, and focal weakness can also be noticed after birth in infants with a stroke [22]. Other neonates may experience subtle or no signs at all, being neurologically normal; therefore, they can be discharged home and misdiagnosed. Asymptomatic patients can especially be those who experience an ischemic event involving the anterior or posterior cerebral artery [23]. In these cases, a neonatal stroke is retrospectively diagnosed later in life when neurological impairment, such as epilepsy and cerebral palsy, occurs [24].

Recently increased access to imaging techniques for newborns has made possible an earlier and easier diagnosis of a neonatal stroke. Performing serial cerebral ultrasound is the first step to identifying cerebral infarcts, while MRI neuroimaging is more appropriate to assess the exact site and extent of the lesion. Patterns of injury may vary from small lesions in the basal ganglia to large ones, involving most of the hemisphere. De Vries et al. established a classification for infarcts that affect the middle cerebral artery by the branch involved: main, cortical, and lenticulostriate [25]. The implication of the main branch or one of the cortical branches is seen more often in term infants, while the involvement of the lenticulostriate branch is more frequently detected in preterm neonates. Regarding the site of the stroke, not only is the main artery affected, but also the watershed regions between arterial fields, with injury mainly due to acute episodes of hypotension.

A thalamic stroke is a particular localization of an ischemic event. Thalamic blood supply is divided into four major thalamic vascular territories supplying particular groups of nuclei. Vascular lesions produce combinations of sensorimotor and behavioral syndromes, depending on which cortical–thalamic connections have been interrupted [26]. Therefore, a thalamic stroke falls into four principal vascular syndromes, with wide clinical and prognosis variations, because even small focal ischemic lesions are seldom confined within nuclear boundaries. The functional properties of different thalamic nuclei are inferred from reciprocal connections with behaviorally defined regions of the cerebral cortex [26].

Following any cerebral ischemic event, there are usually short- or long-term important consequences: cerebral palsy, developmental delays, gross and fine motor impairments and epilepsy, and speech and feeding difficulties.

In a perinatal stroke, the prognosis depends on multiple factors, such as the site and the extent of the lesion evidenced by neuroimaging, abnormal EEG activity, and the presence of factor V Leiden. Several studies have reported an association between arterial stroke location and the prediction of neurological sequelae: left supramarginal gyrus lesions determined speech disorders, left external capsule damage correlated with abnormal gross motor function, and frontal lesions resulted in cognitive impairment [27,28,29,30].

Although the Bayley Scale of Infant Development III is known to be the gold standard tool used to assess neurodevelopment outcomes in the first two years of life in children with a neonatal stroke [31], two early predictors of motor outcomes following a symptomatic perinatal stroke could be effective: the pattern of MRI abnormalities and observation of spontaneous infant movements by a trained observer (General Movement Assessment—GMA) [32].

## 2. Case Report

We present the case of a female child born at 35 weeks of gestation, extracted through the emergency cesarean section for preeclampsia, with a birth weight of 2420 g. This was the first pregnancy of a 23-year-old mother, gravida 1, para 1, with an asymptomatic COVID-19 infection detected in the last trimester at a routine RT-PCR test from a nasopharyngeal swab. Because the patient was asymptomatic with normal blood test results, including the D-dimer value, antiviral medication was not prescribed. After 34 weeks of gestation, the pregnancy became complicated with recurrent headaches and edema. Clinical examination at admittance in the hospital revealed high blood pressure and altered laboratory tests (proteinuria, elevated liver enzymes, and thrombocytopenia) that supported the decision for an emergency cesarean section.

At birth, the child experienced a good fetal–neonatal transition, with an APGAR score of 8 at 1 min and 9 at 5 min, and did not require resuscitation maneuvers due to hemodynamic and respiratory stability, with a normal glycemic state. Postnatal laboratory tests showed normal results, except for a high procalcitonin value, raising suspicion of neonatal infection. Thus, i.v. empiric anti-biotherapy (ampicillin and gentamicin) was started, and a blood culture was sampled.

On the fifth day of life, the newborn presented an apnea episode with generalized cyanosis and decreased muscle tone, remitted after tactile stimulation. The baby was transferred to the NICU for cardio-respiratory monitoring, caffeine citrate was initiated, and antibiotic therapy was escalated to i.v. meropenem and vancomycin. The complete blood count, blood gases, lactate, and biochemical constants were normal, and no glycemic or electrolyte imbalance was noted. Extended metabolic screening showed normal values and the RT-PCR test for SARS-Cov2 was negative. No evidence for toxic or metabolic causes was revealed. Traumatic delivery, birth asphyxia, or polycythemia were also excluded. The blood culture was positive for Group F Streptococcus and the antibiotic therapy was modified, according to the antibiogram, to ceftriaxone and linezolid continued for 14 days with antifungal prophylaxis with fluconazole, until the blood culture was negative.

During NICU hospitalization, the newborn continued to present apneic episodes (1–2 per day), reversible with tactile stimulation, and did not require oxygen therapy or respiratory support, having normal vital signs and arterial blood gases.

The routine cerebral ultrasound that had been performed on the third day of life did not reveal any pathological findings, but on the eighth day of life, the same examination revealed an abnormal hyperechoic round, homogenous, well-differentiated, unilateral image in the left thalamic region, suggestive of a thalamic infarct; it can be observed in Figure 1A,B. Assessing cerebral blood flow using a Doppler examination showed no abnormality.

Performed thrombophilia and coagulation tests were normal. The histopathological examination of the placenta did not provide evidence of ischemic or inflammatory changes.

The baby was referred to a pediatric neurologist who described a normal neurological exam and normal EEG pattern. Echocardiography showed no intracardiac thrombus or other pathology and the abdominal ultrasound was also normal. A cerebral CT scan (native and iv contrast) showed no abnormality at that moment. Subsequent MRI examination acquired with 3T scanners revealed one focal lesion of the left thalamic area, discrete hyperintensity in the T2 sequence without FLAIR suppression, and hypointensity in the T1 sequence with restricted diffusion on ADC and DWI sequences, as shown in Figure 2 and Figure 3. No mass effect was detected. The MRI aspect was consistent with a chronic/subacute ischemic substrate.

From a clinical point of view, after the 12th day of life, apneic episodes ceased under the previously mentioned therapy (no thrombolytic agents or anticoagulant therapy was given); the ongoing neurological examination during hospitalization did not reveal any pathological signs and the baby was discharged home on the 26th day of life. Serial cranial ultrasounds performed until discharge showed the same abnormal, nonprogressive echogenicity in the thalamic area (Figure 4A,B).

At the age of 6 months, the neurological assessment of this patient was normal, showing no signs of sensorial, motor, or cognitive deficits, and the head ultrasound performed at that time showed the same aspect (increased circumscribed echogenicity) of the thalamic lesion. The child will continue a strict follow-up program with an interdisciplinary medical team of pediatricians, pediatric neurologists, and physical therapists.

## 3. Discussion

A neonatal stroke can be easily misdiagnosed, especially when clinical signs are very subtle or can be attributed to other associated pathologies; in most cases, it is even harder to identify the cause that generated the event. Identifying the underlying cause is of paramount importance, as it may minimize the extent of the disease, prevent future recurrence of strokes, and allow early interventions [33]. A recent meta-analysis conducted by Rattani A et al. showed that a third of children with a perinatal stroke developed epilepsy, the size and extent of the lesion being important determinant factors [34]. To our knowledge, two studies suggested an association between the thalamic location of the stroke and cognitive impairment afterward [35,36].

Regarding the etiology of the neonatal stroke that was postnatally discovered in this patient, we found it difficult to summarize it to just one potential factor. We considered the mother’s emergency pathology, preeclampsia, that determined premature birth as a possible determinant cause, among other risk factors, as we previously underlined the probability of a multifactorial etiology.

Although maternal SARS-CoV2 infection did not determine fetal infection, it could have produced an immune-mediated inflammatory reaction that predisposed the neonate to a hypercoagulability state. Previous publications have demonstrated serious placental injury and damaged fetal perfusion linked to the affected coagulation state determined by SARS-CoV2 infection during pregnancy [37,38]. In our case, we cannot confirm the direct association between the cerebrovascular ischemic event and SARS-CoV2 infection; nonetheless, we cannot exclude it as a possible risk factor for cerebral ischemic injuries. Recent studies provide evidence for higher IL-6 cytokine levels in pregnant women with SARS-CoV2 infection than non-pregnant ones; thus, by releasing proinflammatory mediators, the fetus is exposed to a “cytokine storm”, causing consequent cerebrovascular damage [37,38,39].

Early-onset sepsis in this case might have caused the stroke, although data about a septic cerebral embolism in neonates are limited. Few publications have mentioned an association between a septic embolism and neonatal stroke, leading to a hypercoagulable state. Other specific pathogenic mechanisms incriminated during inflammation are: inhibition of antithrombin III, protein C and S, microvascular flow disturbances due to increased release of elastase after activation of granulocytes, and stimulation of the cerebral vascular endothelium by inflammatory cytokines, IL-1 and TNF [40,41,42].

As we mentioned in the Introduction, genetic factors could also be taken into consideration in this patient [43,44], but we were not able to perform genetic tests other than hereditary thrombophilia, which provided normal results. Although the diagnosis of a thalamic stroke is certain and it became clinically significant on the fifth day of life, the exact moment when the ischemic process occurred remains unclear. Ultrasonography is not considered reliable in establishing the timing of injury in the case of ischemic lesions; it may not detect any anomalies by the end of the first week after the event. MRI is much more efficient in detecting early lesions, if available. Typically, ultrasonography shows a cystic evolution of the lesion over the next 4–6 weeks, sometimes with dilatation of the ipsilateral ventricle if the infarction zone is in one cerebral hemisphere [2]. Interestingly, there is a difference in the evolution of the ischemic lesion, a specific destructive pattern depending on the maturity of the brain. The fetal brain has limited capacity for an astrocytic response; therefore, the necrotic tissue is completely reabsorbed, resulting in a fluid-filled cavity (porencephaly), while the mature brain reacts to injury by significant astrocytic proliferation, resulting in lesions containing “soft brain” (astroglial cells). The neonatal brain falls somewhere in between, with a 15% astrocytic response to injury compared to the mature brain. Thus, usually, the postischemic residual lesion evolves to a pure cyst in the second-trimester fetus, to a mostly cystic area with astroglial septa in the last month of gestation and the neonatal period, and to pure astrogliosis with no appreciable cystic component in the mature brain [45]. Our case showed an unusual progression for an ischemic lesion, as after 6 months, the hyperechoic image would have been expected to become a cyst. The localization and a particular reactivity of the surrounding tissues, more similar to a mature brain, may explain this situation.

Doppler ultrasonography in cases of a neonatal stroke can show asymmetrical arterial pulsations, decreased on the affected side in the acute phase and increased in size and several visible vessels in the periphery of the infarct with increased mean blood flow velocity a few days later (so-called “luxury perfusion”) [2]. In the presented case, normal Doppler findings may be due to a late investigation or an older lesion.

No standardized intervention exists for patients with a stroke; therefore, early detection is of crucial importance because there is a critical time window for activity-dependent plasticity to mold corticospinal tract development in the first few years of life. Life support interventions and minimizing the damage from the initial insult seem to be achieved by neuroprotective strategies including hypothermia, growth factors (erythropoietin), antioxidants (melatonin), anti-inflammatory agents (minocycline), or drugs that can reduce excitotoxic damage (topiramate) [32].

Unfortunately, there are often significant delays between the ischemic event, onset of any symptom, and final diagnosis of a perinatal stroke. In total, 40% of presumed perinatal strokes are detected outside the neonatal period. The mean age for the diagnosis of a perinatal stroke is 12.6 months [46].

New interventions that could potentially reduce morbidity after ischemic insults are in the stage of clinical trials; for example, therapy with autologous umbilical cord blood-derived stem cells seems to contribute to the limitation of the infarcted zone and a better functional outcome [32]. It has been recently demonstrated that astrocytes may play a protective role after a stroke by the production of growth factors and regulating glutamate homeostasis [6]; therefore, strategies that target reactive astrocytes for stroke recovery are subject to present research. Translation of the actual results into effective clinical therapy needs time and further studies [6]. Modulating cortical excitability using noninvasive brain stimulation with transcranial direct current stimulation or using repetitive transcranial magnetic stimulation are tested in adults for balancing excitability in the affected cortex, but it has not yet been experimented with in the immature brain [32].

For children with already established cerebral palsy, new interventions such as constraint-induced movement therapy (CIMT) or exploitation of the “mirror neuron system” are under investigation to improve motor outcomes after a stroke [47].

Advances in DNA sequencing, genetic analyses, and the development of new tools to correct human gene mutations may represent in coming years a future therapy opportunity to prevent a stroke in patients with genetic predisposition [13].

We consider the case we presented extremely challenging as the patient was diagnosed with early-onset sepsis with uncommon etiology (group F streptococcus), associated with a perinatal stroke with uncommon localization, both severe conditions, but with poor symptomatology. It is unclear if the infectious disease caused the ischemic process, or if the two pathologies simply co-existed in this same patient. An important additional risk factor could be maternal SARS-CoV2 infection during pregnancy. Early diagnostics and interventions have made possible a good disease course and a satisfactory outcome up until now.

## 4. Conclusions

A neonatal stroke is an important cause of long-term neurological and cognitive impairments, including cerebral palsy, epilepsy, hemiplegia, language and behavior disorders, and developmental delay [27]. Unfortunately, clinical experience shows that a neonatal stroke can be encountered in patients without known related family history, diagnosed pregnancy pathologies, or acute perinatal injuries, and clinical signs may be subtle and nonspecific. We suggest that routine head ultrasound examinations should be performed for patients at risk.

We recommend that physical, speech, and occupational rehabilitation, as well as other therapies, be initiated as soon as possible during the first two years of life when the brain has amazing plasticity. Early interventions could significantly reduce motor and cognitive deficits after a neonatal stroke.

The case we presented intends to emphasize the need for thorough investigations of newborns delivered from mothers with infectious pathology, bacterial or viral, as this could have a significant impact on the neonatal outcome. Pandemic experience with SARS-CoV2 infections during pregnancy warrants close monitoring, both perinatal and long-term, to discover and intervene in due time any potential risk and improve the neurodevelopmental outcome.

Undoubtedly, early identification tools for neurodevelopmental deficits, such as the General Movement Assessment, should be introduced in daily clinical practice and further studies regarding the use of erythropoietin, melatonin, or other neuroprotective agents would be helpful to obtain the best possible recovery after perinatal neurological impairments.

## Figures and Tables

**Figure 1 children-10-00958-f001:**
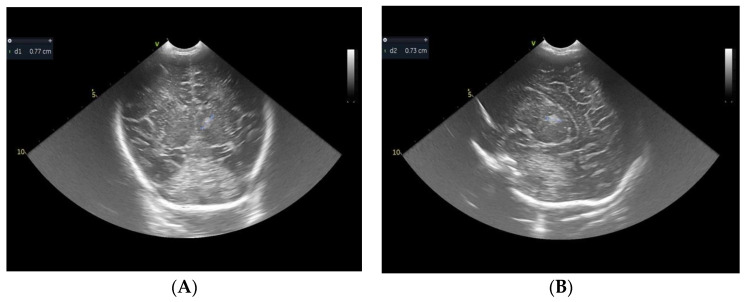
Cranial ultrasound, coronal (**A**) and sagittal (**B**) plane, shows a left-sided increased echogenicity of thalami.

**Figure 2 children-10-00958-f002:**
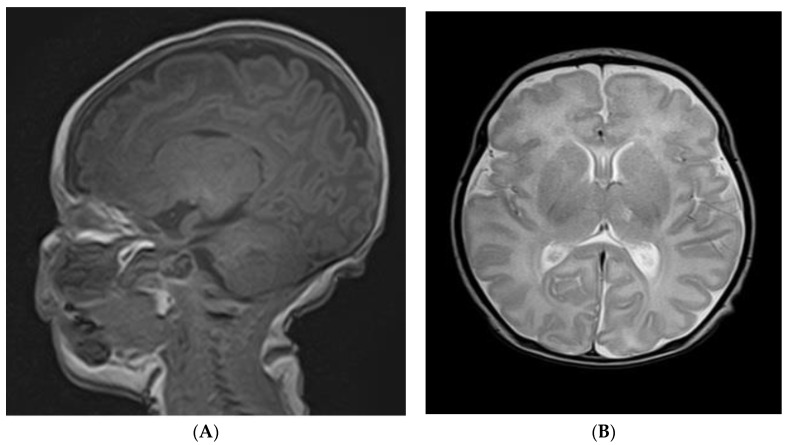
(**A**,**B**) MRI sequence confirmed thalamic lesion of 5/3 mm.

**Figure 3 children-10-00958-f003:**
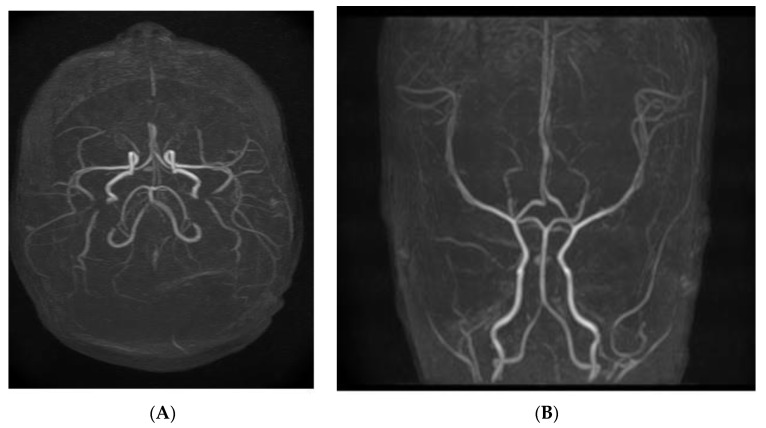
(**A**,**B**) Normal vascular pattern on MRI.

**Figure 4 children-10-00958-f004:**
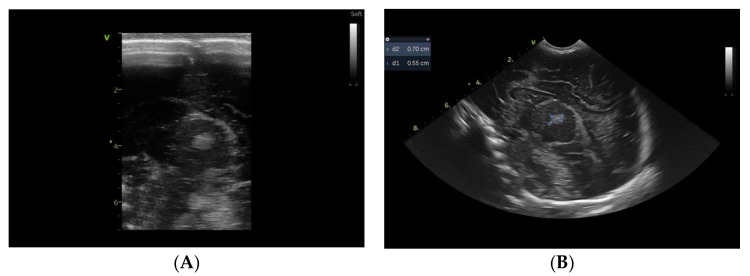
Cranial ultrasound, sagittal plane, linear probe (**A**), and convex probe (**B**), performed at discharge shows the same increased echogenicity in the thalami.

## Data Availability

No new data were created or analyzed in this study. Data sharing is not applicable to this article.

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
