# Peer review of "SARS-CoV-2 Infection during Pregnancy Followed by Thalamic Neonatal Stroke—Case Report"

_children, 2023, doi:10.3390/children10060958_

Round 1

Reviewer 1 Report

Thank you for the opportunity to read this very interesting manuscript with a unique presentation, as the mother exposed the fetus to Covid in-utero. You have done an amazing job introducing the topic, and providing the reader with detailed presenting and diagnostic information. You also added valuable additional information and citations in the discussion. Good luck!

Other than a few minor typos the English is fine.

Author Response

Please find attached our reply. Thank you!

Reviewer 2 Report

SARS-CoV-2 infection during pregnancy followed by thalamic 2 neonatal stroke– case report.

In this manuscript, the authors report a case of a late preterm infant diagnosed with a thalamic stroke in the 5th day of life, with no clinical signs except for except for repeated episodes of apnea. The mother had SARS CoV2 infection late in pregnancy and the infant suffered an early bacterial neonatal sepsis. The authors conclude that early identification and physician awareness of this condition in the neonatal period are of paramount importance to reduce developmental postischemic damage.

Comments and suggestions.

Abstract.

The final sentence “Early identification and awareness of the physicians about this condition in neonatal period have paramount importance in order to reduce the developmental postischemic damage”, seems grammatically incorrect. Please, re-phrase conveniently.

Introduction.

The introduction is a bit long and too general. It should be more focused on the case.

References must be numbered in order of appearance in the text, and before the punctuation signs.

Page 1, lines 41-42, it reads “in a study conducted by Berfelo et al. the straight sinus was most commonly affected and an associated thalamic hemorrhage was common. [32]”, but ref 32 do not correspond to Berfelo et al. Please, clarify or correct it.

Page 2, line 60-61, “a population-based- case-control study of 1997- 2002 births at Kaiser Permanente Northern California” is mentioned. Please, give the appropriate reference.

In addition, the whole paragraph seem to refer to genetic studies, but the reference given (num. 38) corresponds to an image study. Is that correct?

Line 79. Reference 39 seems to be wrong. Did you meant 38?

Line 80. “Born from mothers…” Should be “born to mothers”.

Line 82-83: “Hypotesys”, should be hypothesis.

Line 89. Ref. 34 should be 33.

Line 95. “This cases” should be “these cases”.

Line 98. “Imagistic techniques”, should be “Imaging techniques”.

Etc. And many more.... English must be fully revised.

Case report.

Line 114. Which “altered laboratory test”? Please, specify.

Figures titles and legends should be individual. If grouped, name them Figure 1 A and Figure 1 B, for example.

Discussion.

Once again, the discussion is too long, too general, and quite unfocused. Please try to focus the discussion on the specifics of your case, if any.

References.

The format of the references seem to be heterogeneous. Please, uniform it according to the guidelines. There are also many typos to correct.

Line 115, 275, 278, and more…. Reference [41] is mentioned, but it is lacking in the references section. 

Line 350. Reference 13. The title of the paper is lacking..

More typos.

Please, review and correct all typos. There are many through the text.

For instance, page 1 line 16: it reads “blood viscosityand risk of dehydratonduring the first few days of life”. 

Line 20 “a lot” is not an appropriate scientific language. Please, consider “many” instead.

Etc….

The manuscript needs to be extensively reviewed by a native English speaker.

Author Response

(The authors gave the same response as above.)

Reviewer 3 Report

Dear Authors

Thank you for the opportunity to review this interesting research paper.

Some spacing inconsistencies in abstract

line 16 make new sentence from 'but'

line 21 removed one of the 'except for'

questioning the numberng - though they were supposed to be consecutive and not go 1, 32, 2,3,4,33

line 41 start new sentence and put in a linking word, maybe 'incontrast'

line 48 needs a link from previos point. Put refernces after first sentence

line 49 - first what

line 54 - link between points needed

line 59 - undertaken between 1997 and 2022 at Kaiser Permanent Northern California, identified 13 infants ...

make this two sentences as it is too long and unclear

put reference at the end of first sentence about this study

line 72 sentence too long and not clear and needs referncing

line 76 reference

line 77 needs a link between these two points

line 80 link between points needed

line 85 ? should not have initial here and put reference at end of sentence. sentence not clear and spacing issue

line 93 - new sentence from therefore

line 94 what is 'this' referring to

line 101 - what is 'this' referring to

line 109 paragraph needs referencing - wonder if this should be earlier rather then in between two diagnostic discussion paragraphs

line 115 re prognosis shuold be at end of previous paragraph

this section needs rearranging so that all of the diagnostic discussion is together

a paragraph is more then two sentences - this needs addressing here

last three paragraph do not flow well here - suggest rearranging this section and put links in between points. Rather disjointed and difficult to follow.

what is significance of line 133 sentence - conclude

case report - mention that this is the first child twice

not sure how this can be an uneventful pregnancy is she had c secton for preeclampsia

line 141 - because the patient

line 147 - did not not didn't

line 166 - routine, did not

refer to figure in text

line 179 sentence not clear

line 180 ? referred

line 182 missing full stop

line 195 ? no thrombolithic agents

line 200 ? 'this patient'

line 207 'it is'

line 211 ? should not have initial

line 224 needs reference

line 231 not clear and possible a word missing. Whole sentence is not clear and too long

line 240 ? reference and not a paragraph

line 242 - became clinically significant

line 261 spello here making sentence unclear

line 277 ? perinatal stroke

line 278 time period not clear

line 279 onwards - ? combine these paragraphs up to line 291 maybe

conclusion should have recommendation

general issue of sentence construction, typos, spacing and paragraph structure needs reviewing.

Poor English 

Author Response

(The authors gave the same response as above.)

Round 2

Reviewer 2 Report

Thank you for the review. The manuscript has improved significantly. Just some minor points.

Line 48. Please write references in order: [2,9] instead of [9,2].

Line 51. When several references are continuous, such as [10,11,12], is better to write [10-12].

Line 80. The same for [18,19,20]. Better [18-20].

Line 124. [27-30], instead of [27,28,29,30].

Line 227. [37-39], instead of [37,38,39].

Line 234. [40-42], instead of [40,41,42].

References 45,46, and 47 are mentioned in the text, lines 254, 273, and 287, but they are not in the references section. Please, add them.